# Investigating Factors for Travelers' Parking Behavior Intentions in Changchun, China, under the Influence of Smart Parking Systems

**Yunxiang Zhang, Xianmin Song, Pengfei Tao \*, Haitao Li, Tianshu Zhan and Qian Cao**

Nanling Campus, Jilin University, 5988 Renmin Dajie, Changchun 130022, China;
zhangyx21@mails.jlu.edu.cn (Y.Z.); songxm@jlu.edu.cn (X.S.); lihait@jlu.edu.cn (H.L.);
zhants21@mails.jlu.edu.cn (T.Z.); caoqian20@mails.jlu.edu.cn (Q.C.)
\* Correspondence: taopengfei@jlu.edu.cn

**Abstract:** Unraveling the determinants of travelers' parking behavior intentions is critical to the widespread adoption of smart parking systems (SPSs), which hold the promise of greatly enhancing parking efficiency and optimizing resource allocation within urban spaces. Our study pioneers the use of an integrated methodology combining structural equation modeling (SEM) and hierarchical regression modeling (HRM) to dissect the complex interplay of these determinants. We found that, in the structural equation model, social influence notably stood out as having the most significant impact on the intention to utilize SPSs. Notably, while perceived privacy concerns may have ranked lower in terms of influence among these factors, their role was relatively crucial, particularly given the contemporary emphasis on data security. Moreover, within the hierarchical regression model, driving experience was found to play a crucial role in determining the intention to use SPSs. Equally important, our research revealed a divergence in parking intentions between individuals with children and those without. This points towards the imperative need for personalized strategies that can cater to the diverse requirements of different user demographics. This research offers guidance for operators of SPSs aiming to formulate targeted approaches.

**Keywords:** smart parking systems; parking behavior intentions; influential factors; structural equation model; hierarchical regression model



## 1. Introduction

Escalating global urbanization has brought a critical challenge to the forefront—understanding and shaping travelers' behavior to develop effective transportation policies [1]. Of paramount importance among these behaviors is parking, as it directly impacts the efficiency of urban transportation systems [2]. Driven by this urgency, our research sought to decipher the behavioral intricacies of parking in today's evolving urban landscapes.

The swift growth of urban populations has significantly intensified the demand for parking spaces, spawning numerous issues such as illegal parking, traffic congestion, and increased search durations for parking spots. As a promising solution to these challenges, SPSs have emerged, leveraging cutting-edge technologies to provide real-time availability, guide drivers to open spots, and streamline payment procedures [3]. By optimizing the utilization of parking spaces, SPSs can significantly reduce instances of illegal parking, thereby enhancing traffic flow and overall urban mobility [4,5].

Despite the promising benefits of SPSs, their widespread adoption has been hindered by a substantial knowledge gap concerning travelers' intentions in relation to using these systems. This gap often results in apprehension amongst potential users about embracing such novel technologies [6]. Thus, our research aims to fill this knowledge gap and is pioneering in its endeavor to offer comprehensive insights into user attitudes and

preferences towards SPSs. Specifically, we aspire to shed light on the practical determinants that would encourage the acceptance and usage of SPSs, contributing to strategies that would promote their adoption, which is crucial for the improvement of urban mobility and the alleviation of urban traffic issues.

Departing significantly from traditional parking behavior research, SPSs introduce fresh considerations, including technology acceptance, incentive mechanisms, and privacy concerns [7]. These aspects necessitate an innovative approach to understanding travelers' parking behavior intentions, which underscores the uniqueness of our study. Our research goes beyond examining standard parking factors like pricing and availability, focusing instead on the acceptance and adoption of novel technologies such as smartphone applications and automated parking guidance systems [8]. We delve into the realm of online mobile payments and explore how incentive mechanisms, like parking coupons or time-limited free parking, can be effectively implemented online. Moreover, we address the critical issue of privacy concerns arising from data sharing between personal devices and platforms.

Given the distinct nature of SPSs, traditional research methods may not adequately capture the complexities of parking behavior intentions. To rectify this, our study adopts an innovative and comprehensive approach, analyzing travelers' parking behavior intentions within Changchun's SPS using a combination of a hierarchical regression model and a structural equation model. This dual-pronged approach considers both psychological and objective factors influencing travelers' intentions, providing a comprehensive understanding of SPS adoption.

The remainder of this article is structured as follows: Section 2 presents a review of the pertinent literature. In Section 3, we explore travelers' intentions in relation to the use of SPSs, detailing the conceptual framework and data collection process, along with the sample profile. Section 4 delves into the methods and results, including an examination of the factors influencing smart parking adoption using hierarchical regression and analyses of objective factors, the goodness of fit, and the estimated results. Furthermore, we investigate psychological factors in smart parking system adoption, test several hypotheses, and analyze direct, indirect, and total effects. Finally, Section 5 concludes the paper, summarizing key findings.

## 2. Literature Review

Examining travelers' parking intentions offers crucial insights into the perceived attractiveness of SPSs. Past research has predominantly focused on key determinants affecting parking intentions, such as accessibility [9–12], parking convenience [13–15], parking cost [16,17], and parking availability [18–21].

In addition to these primary factors, research has also highlighted latent variables, which are not directly observable, such as safety [22,23], privacy [24–26], attitudes [27,28], and social influence [29–33]. These latent variables are grounded in well-established theoretical frameworks, such as the theory of planned behavior (TPB), the technology acceptance model (TAM), and privacy calculus theory (PCT), which provide a solid foundation for exploring the relationships between these latent factors and parking intentions.

The TPB is a widely accepted model for intentions. The TPB comprises three primary components (attitudes, subjective norms, and perceived behavioral control), which collectively influence behavioral intentions and, ultimately, actual behavior. Researchers have applied the TPB to study parking intentions, emphasizing the effect of social norms on usage frequency [34–37]. These researchers discovered that both subjective norms (actions of important others) and descriptive norms (actions of the majority) significantly influence parking behavior.

The TAM is a widely employed framework for users' acceptance of new technologies. The TAM is a theoretical model that explains users' acceptance of new technologies based on perceived usefulness, perceived ease of use, and perceived trust. Numerous studies have applied the TAM to investigate parking intentions, focusing on the adoption of smart

parking technologies [11,18,29]. They found that addressing perceived usefulness, ease of use, and trust positively impacted user acceptance, leading to increased adoption rates for smart parking solutions.

PCT posits that individuals weigh the potential benefits of disclosing their personal information against the potential risks to their privacy [38,39]. This theory has been applied in various contexts, including mobile applications [40,41], online information privacy [42], and mobile hotel booking loyalty [43,44]. SPSs rely on mobile applications and online information.

To capture the intricate relationships between factors and parking intentions, researchers have utilized various data collection and analytical methods, including questionnaires, surveys, interviews, observational studies, choice experiments, revealed preference studies, and agent-based simulations [9,12,14,15,45,46]. Statistical analyses, such as regression models, SEM, and cluster analyses, have been employed to determine the associations between these factors and parking intentions [18,47,48].

Although previous research has contributed significantly to our understanding of parking intentions, certain gaps remain, particularly in the context of SPSs. Existing studies have primarily emphasized objective factors, such as accessibility, convenience, and cost, while more nuanced perceptual elements, including privacy concerns and technology acceptance in relation to parking intentions, have received limited attention. This lack of focus on perceptual factors may have led to an incomplete understanding of travelers' parking behavior and the factors that drive their decisions. The present study aims to address these gaps by focusing on the interplay of both objective and perceptual factors, offering a more holistic view of travelers' parking intentions.

## 3. Investigating Travelers' Intentions to Use SPSs

### 3.1. Conceptual Framework

The conceptual framework for this study, as illustrated in Figure 1, is segmented into two core components: the relationship between objective factors and the intention to use SPSs and the connection between psychological factors and the intention to use SPSs. The latter domain, the focus of this explanation, scrutinizes attitudes towards smart parking, perceived usefulness, perceived ease of use, social influence, and privacy concerns, all of which interact to exert direct or indirect effects on the intention to use SPSs.

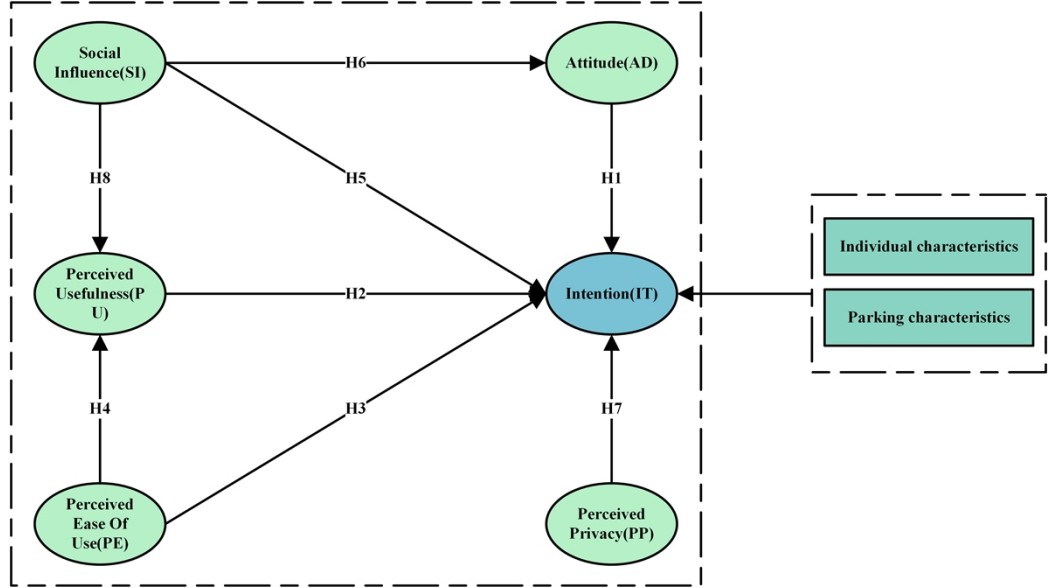

**Figure 1.** Schematic representation of intentions in the context of SPSs.

Psychological factors, in the context of this study, refer to the internal cognitive and affective elements that underpin an individual's decision-making process with regard to using SPSs. These elements encompass a spectrum of individual determinants, such as beliefs, attitudes, perceptions, motivations, and emotions. We probe these specific psychological variables—attitude, perceived usefulness, perceived ease of use, social influence, and privacy concerns—because they are firmly grounded in well-established theoretical models in the fields of technology acceptance and behavior prediction, like the TPB [49], the TAM [50], and PCT [39].

The choice to adopt the TPB, the TAM, and PCT as the theoretical compass for this study was an informed and deliberate one. For instance, the TPB accentuates the importance of attitudes, subjective norms, and perceived behavioral control in shaping behavioral intentions, providing a suitable framework to investigate users' attitudes towards SPSs and the social influences impacting their intention to use SPSs. The TAM predicates that perceived usefulness and ease of use are essential determinants in technology acceptance and usage. Given SPSs' innovative nature and the paradigm shift they introduce with regard to traditional parking methods, understanding how users perceive their utility and user-friendliness is vital. The PCT posits that individuals weigh the potential benefits against the possible risks when disclosing personal information, making it an appropriate model to investigate privacy concerns associated with SPSs, which often involve the collection and use of personal data. The individual psychological variables are listed in Table 1.

**Table 1.** Key variables used in assessing SPSs acceptance.

| Variables | Description |
|---|---|
| Attitude | Representing travelers' overall assessment of SPSs, this variable originates from the TPB. It underscores the role of attitudes in shaping behavioral intentions and actions. |
| Perceived usefulness and perceived ease of use | These critical determinants of technology acceptance reflect the users' beliefs that SPSs can enhance performance (usefulness) and are user-friendly (ease of use). Rooted in the TAM, these concepts propose that users' adoption inclination increases if the technology is perceived as useful and easy to use |
| Social influence | This factor, based on the TPB, measures the extent to which individuals believe their social network endorses or utilizes SPSs. This perception underpins the TPB's emphasis on the role of individual attitudes and subjective norms in determining behavioral intentions and actual behaviors; |
| Privacy concerns | Grounded in the PCT, this aspect captures individuals' apprehensions about person-al information disclosure. In the SPS setting, concerns may arise around the collection and utilization of personal details by the system or third parties |
| Intention | Also based on the TPB, intention indicates that an individual's behavioral intention is a product of their attitudes, subjective norms, and perceived behavioral control. In the context of SPSs, intention gauges the likelihood of travelers adopting and using the technology, premised on their perceptions of its usefulness, ease of use, and trustworthiness, as well as social influence and privacy concerns. |

From this structural model of travelers' intentions to use SPSs, the following hypotheses can be proposed:

**H1.** *A more positive attitude towards SPSs will result in a higher intention to use them;*

**H2.** *SPSs' perceived usefulness will positively affect travelers' intention to use them;*

**H3.** *SPSs' perceived ease of use will positively impact travelers' intention to use them;*

**H4.** *SPSs' perceived ease of use will positively influence their perceived usefulness;*

**H5.** *Social influence will have a significant positive effect on travelers' intention to use SPSs;*

**H6.** *Social influence will have a significant positive effect on attitudes towards SPSs;*

**H7.** *Privacy concerns will have a negative effect on travelers' intention to use SPSs;*

**H8.** *The perceived usefulness of SPSs will be positively affected by social influence.*

*3.2. Data Collection and Sample Profile*

Our study unfolds within the dynamic urban landscape of Changchun, a vital hub in northeast China. As the most populous and economically vibrant city in Jilin province, Changchun is a bedrock of industrial activity, a pivotal point in the regional transportation network, and a vibrant hub of cultural exchange. We focus our attention on a region characterized by its intense parking requirements, especially during peak hours. This sector, a key intersection in the city's transportation matrix, is a potential ground zero where the strategic deployment of smart parking could catalyze transformative impacts on traffic flow radiating to neighboring zones. Thus, our empirical focus lies within the prominent parking areas ensconced within the bustling perimeter of Changchun city's Hongqi Street central business district (CBD).

To delve into this urban phenomenon, we deployed a multimodal investigative approach, intertwining both face-to-face and digital channels of survey distribution. Spanning a three-month period from July to September 2022, our surveying efforts were targeted at individuals aged 18 or over residing within the immediate vicinity of the study area and with prior exposure to intelligent parking systems. The face-to-face questionnaires were administered during peak parking demand hours (morning: 8:00–10:00, evening: 18:00–20:00) across both weekdays and weekends, thereby ensuring a comprehensive appraisal of user behaviors and requirements within the area. Out of the 270 questionnaires disseminated amongst Changchun's urban commuters, we received 219 completed responses, translating to an impressive 81.1% response rate.

Our research questionnaire was designed in three distinct sections. The initial segment dedicated to the demographic data of the participants is represented in Figure 2. In our dataset, we found that males represented 53.4% of the respondents, with females comprising the remaining 46.6%. This near-equitable gender distribution was deliberately designed to ensure a balanced investigation into the intent to utilize intelligent parking systems. The age distribution, which closely resembles a normal curve, mostly encompassed drivers aged between 36 and 45 years. It was also observed that drivers with over seven years of experience were more common. Respondents with high academic achievement, notably those holding a bachelor's degree or above, accounted for a significant 54.8% of the sample. Remarkably, a substantial portion of the sample (83.1%) consisted of individuals with children at home, indicating a potential need for efficient parking solutions among family-oriented individuals. The second part of the questionnaire encompassed a series of propositions concerning psychological aspects. These included behavioral intentions, attitudes, perceived usefulness, ease of use, social influence, and privacy concerns pertaining to SPSs. The questionnaire used a five-point Likert scale ranging from 1 (strongly disagree) to 5 (strongly agree). The third part of the questionnaire concerned information on parking characteristics. This information included time spent searching for parking, parking pricing, the distance from the destination, parking coupons, and time-limited free parking.

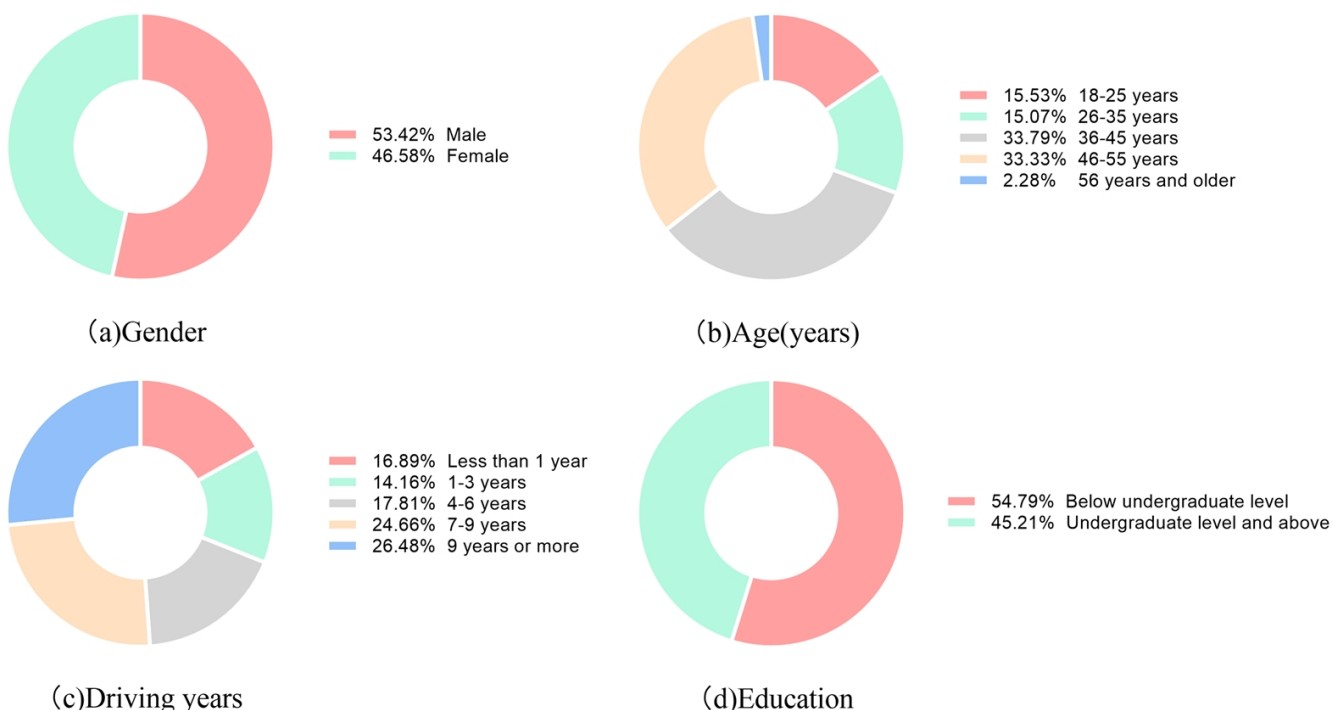

**Figure 2.** Comprehensive breakdown of demographic characteristics.

To meticulously operationalize variables such as intention, attitude, privacy concerns, and social influence in the structural model, it was imperative to leverage multiple observed indicator variables. These observed indicators provided measurable dimensions to the underlying latent constructs, helping to effectively evaluate their influence on parking behavior intentions.

Table 2 provides a detailed inventory of observed indicator variables corresponding to each latent variable. It further includes descriptive statistics, such as the mean, standard deviation (St. Dev.), skewness, and kurtosis, for each observed indicator. The mean and standard deviation help summarize the central tendency and dispersion of the variables. In contrast, skewness and kurtosis provide insights into the shape and tails of the variable distribution, shedding light on the data's asymmetry and tail-heaviness.

Figure 3 offers a comprehensive visual representation of the score distributions for each latent variable, which range from 1 to 5. Within each bar chart, different color-coded sections represent the score distributions for the corresponding observed indicator variables. This dual-level representation allows us to examine the overall distribution of latent variable scores and the individual contributions of each observed indicator. This visualization assists in understanding the spread and concentration of the data points, giving a clear idea about the respondents' collective responses and their alignment with the measured constructs.

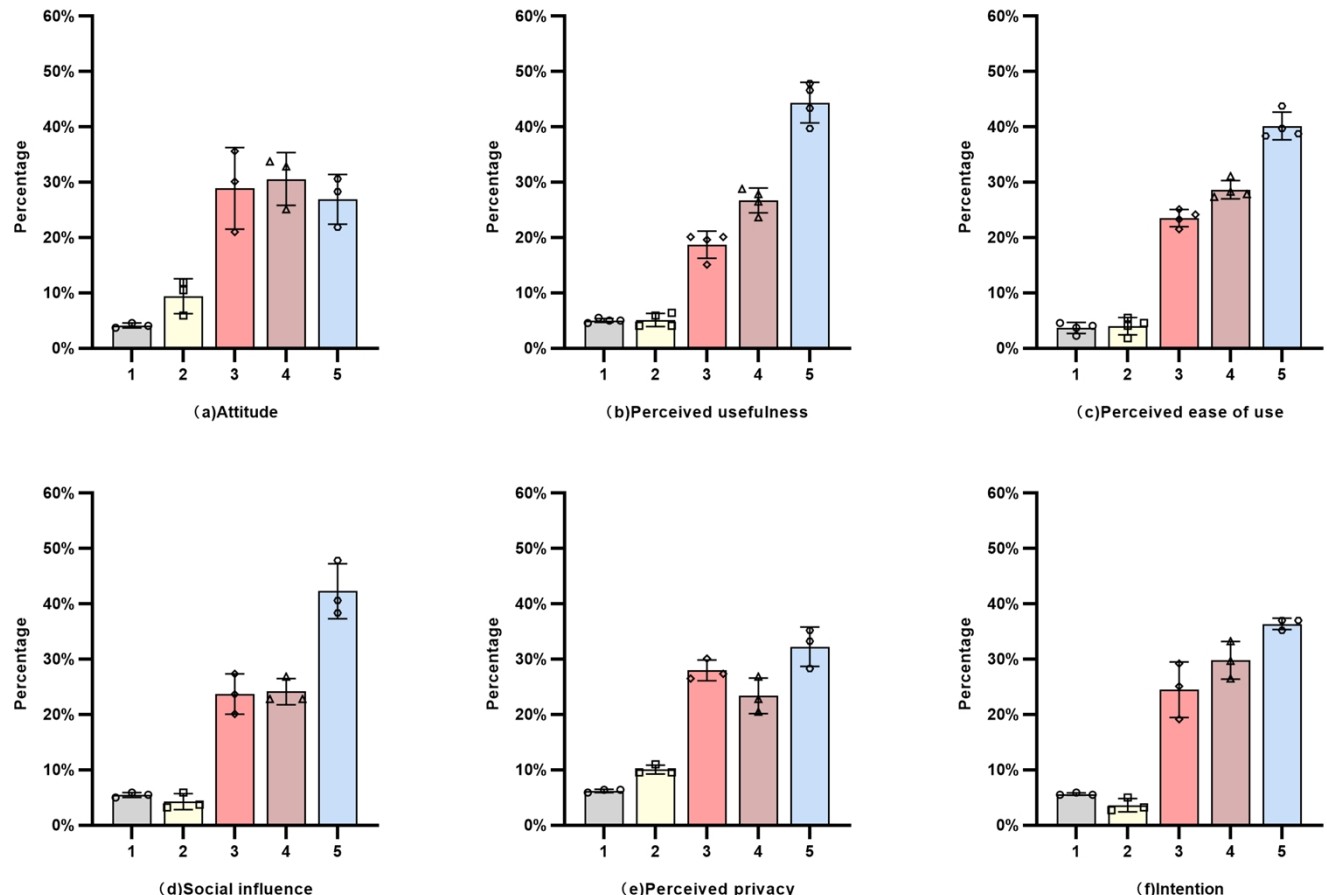

**Figure 3.** Distribution profiles of psychological constructs in the dataset.

**Table 2.** Descriptive statistics and observed indicator variables for the latent variables.

| Latent Variable | Observed Indicator Variable | Mean | St. Dev. | Skewness | Kurtosis |
|---|---|---|---|---|---|
| Attitude (AD) | AD1: Using smart parking system is a wise decision | 3.61 | 1.15 | −0.393 | −0.67 |
| | AD2: My experience with the smart parking system was pleasant | 3.76 | 1.12 | −0.685 | −0.292 |
| | AD3: Overall, I am satisfied with the smart parking system | 3.63 | 1.006 | −0.442 | −0.012 |
| Perceived usefulness (PU) | PU1: Smart parking service fees save waiting time | 4.1 | 1.115 | −1.253 | 0.931 |
| | PU2: Smart parking service fees save travel time | 4.02 | 1.153 | −1.068 | 0.39 |
| | PU3: Smart parking service fees simplify the search for parking spaces | 3.98 | 1.133 | −0.976 | 0.198 |
| | PU4: Smart parking service fees save walking time from car parks to destinations | 3.92 | 1.142 | −0.918 | 0.106 |
| Perceived ease of use (PE) | PE1: I can quickly locate a smart car park | 3.91 | 1.099 | −0.852 | 0.177 |
| | PE2: I can quickly learn to pay for smart parking | 4.06 | 1.037 | −1.017 | 0.684 |
| | PE3: Smart stop charge model is more convenient than the conventional one | 4 | 1.02 | −0.846 | 0.153 |
| | PE4: Smart parking fees easier than conventional parking | 3.93 | 1.088 | −0.856 | 0.172 |

**Table 2.** *Cont.*

| Latent Variable | Observed Indicator Variable | Mean | St. Dev. | Skewness | Kurtosis |
|---|---|---|---|---|---|
| Social influence (S1) | S1I: Smart parking fees reduce scrambled parking | 3.89 | 1.156 | −0.845 | 0.045 |
| | SI2: Smart stop service charges help reduce traffic congestion | 3.88 | 1.141 | −0.822 | −0.037 |
| | SI3: Smart parking service fees take full advantage of parking resources | 4.04 | 1.151 | −1.102 | 0.47 |
| Perceived privacy (PP) | PP1: Payment of smart parking service fees may divulge personal information | 3.65 | 1.215 | −0.49 | −0.664 |
| | PP2: Smart parking facilities may disclose license numbers due to imperfect services | 3.6 | 1.19 | −0.499 | −0.588 |
| | PP3: Payment of smart parking service fees may divulge location data | 3.72 | 1.209 | −0.589 | −0.578 |
| Intention (IT) | IT1: I would like to continue using smart parking services in the future | 3.9 | 1.104 | −0.914 | 0.388 |
| | IT2: I would recommend smart parking services to friends and relatives | 3.82 | 1.131 | −0.769 | 0.059 |
| | IT3: Hope smart parking fees become widespread in more areas | 3.91 | 1.12 | −0.995 | 0.419 |

## 4. Methods and Results

### *4.1. Examining Factors Influencing Smart Parking Adoption with Hierarchical Regression*

4.1.1. Analyzing Objective Factors in Smart Parking Adoption

The hierarchical regression model was employed to investigate travelers' smart parking choice behavior as it allows for an in-depth examination of the individual and combined effects of various factors. The model has three layers.

The first layer of the model focuses on personal characteristics, providing insights into how demographic factors influence parking behavior. For instance, it may reveal that younger drivers are more inclined towards smart parking or that families with children have a higher propensity for utilizing intelligent parking facilities. These findings can offer valuable information for devising parking policies and infrastructure tailored to the diverse needs of different user groups.

In the second layer, the model investigates the impact of external factors, such as parking costs, distance to the destination, and parking space availability. Understanding how these factors affect travelers' parking behavior can assist in formulating smart parking policies that closely align with local conditions and strike a better balance between the demands of smart parking and traditional parking.

The third layer explores the influence of incentive mechanisms on travelers' parking behavior. By evaluating the effectiveness of various incentive mechanisms (e.g., time-limited free parking, coupon incentives), policymakers can identify the most efficient approaches to promote sustainable and efficient parking options.

The independent variables posited in the study were: gender (X1), age (X2), education (X3), presence of children in the family (X4), driving experience (X5), berthing time (X6), parking price (X7), distance to destination (X8), smart parking coupons (X9), and time-limited free parking (X10).

The model equations incorporating these relationships were as follows.

Let $Y$ denote the dependent variable (response variable), and let $X_i$ represent the independent variables (explanatory variables) for $i = 1, 2, \ldots, 10$. We define the following symbols:

$\alpha_j$—intercept for the $j$-th model, $j = 1, 2, 3$;

$\beta_{ij}$—coefficient of the $i$-th independent variable in the $j$-th model, $i = 1, 2, \ldots, 10$; $j = 1, 2, 3$;

$e_j$—error term for the $j$-th model.

The hierarchical regression model consisted of the following three models:
Model one (base model):

$$Y = \alpha\_1 + \sum\_{i=1}^{5} (\beta\_1i \times X\_i) + e\_1$$

Model two (incorporating additional independent variables):

$$Y = \alpha\_2 + \sum\_{i=1}^{8} (\beta\_2i \times X\_i) + e\_2$$

Model three (incorporating further independent variables):

$$Y = \alpha\_3 + \sum\_{i=1}^{10} (\beta\_3i \times X\_i) + e\_3$$

In these equations, $\alpha\_j$ coefficients represent intercepts, $\beta\_{ij}$ coefficients signify the impact of independent variables on the dependent variable, and $e\_j$ denotes error terms. These three models form the three tiers of the hierarchical regression model, allowing for the gradual inclusion of more independent variables to analyze their influence on the dependent variable.

### 4.1.2. Goodness of Fit and Estimated Results

Figure 4 presents the outcomes of the three models (model one, model two, and model three) employed to examine the factors influencing the decision to utilize SPSs. The standardized coefficients ($\beta$), standard errors (SE), and regression coefficients (B) are provided in the results.

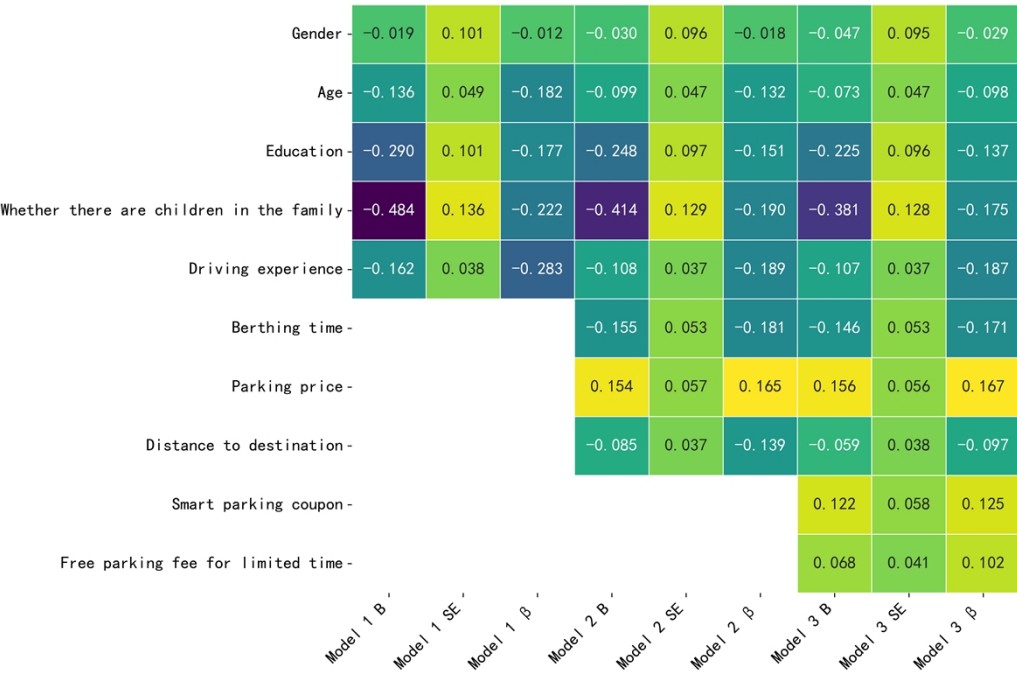

**Figure 4.** Heatmap representation of hierarchical regression analysis outcomes. Generally, the colors used in a heatmap figure represent different ranges of values, transitioning from a lighter to darker hue, or vice versa, to indicate increasing or decreasing values respectively.

Model one incorporated five variables: gender, age, education, presence of children in the family, and driving experience. The results demonstrate that education ($\beta = -0.177$, $p < 0.01$), age ($\beta = -0.182$, $p < 0.01$), presence of children in the family ($\beta = -0.222$, $p < 0.001$), and driving experience ($\beta = -0.283$, $p < 0.001$) significantly impact the choice to use SPSs. The $R^2$ value was 0.209, indicating that the variables accounted for 20.9% of the variance in the dependent variable.

Model two introduced three additional variables: berthing time, parking price, and distance to destination. The outcomes reveal that education (β = −0.137, *p* < 0.05), age (β = −0.132, *p* < 0.05), presence of children in the family (β = −0.175, *p* < 0.01), driving experience (β = −0.187, *p* < 0.01), parking price (β = 0.165, *p* < 0.01), berthing time (β = −0.181, *p* < 0.01), and distance to destination (β = −0.097, *p* < 0.05) significantly influence the choice to use SPSs. The $R^2$ value increased to 0.303, signifying that the variables accounted for 30.3% of the variance in the dependent variable. The $R^2$ change value was 0.277, indicating that the additional variables in model two substantially enhanced the model's predictive capability.

Model three integrated two more variables: smart parking coupons and time-limited free parking. The findings suggest that education (β = −0.225, *p* < 0.01), presence of children in the family (β = −0.381, *p* < 0.001), driving experience (β = −0.107, *p* < 0.01), berthing time (β = −0.146, *p* < 0.01), smart parking coupons (β = 0.125, *p* < 0.05), and parking price (β = 0.167, *p* < 0.01) significantly impact the decision to use SPSs. The $R^2$ value increased to 0.331, signifying that the variables accounted for 33.1% of the variance in the dependent variable. The $R^2$ change value was 0.298, implying that the additional variables in model three considerably improved the model's predictive capability, as shown in Table 3.

**Table 3.** Hierarchical regression model results for different variables.

| Item | Model One | | | Model Two | | | Model Three | | |
|---|---|---|---|---|---|---|---|---|---|
| | B | SE | β | B | SE | β | B | SE | β |
| Gender | −0.019 | 0.101 | −0.012 | −0.03 | 0.096 | −0.018 | −0.047 | 0.095 | −0.029 |
| Age | −0.136 | 0.049 | −0.182 ** | −0.099 | 0.047 | −0.132 * | −0.073 | 0.047 | −0.098 |
| Education | −0.29 | 0.101 | −0.177 ** | −0.248 | 0.097 | −0.151 * | −0.225 | 0.096 | −0.137 * |
| Whether there are children in the family | −0.484 | 0.136 | −0.222 *** | −0.414 | 0.129 | −0.19 ** | −0.381 | 0.128 | −0.175 ** |
| Driving experience | −0.162 | 0.038 | −0.283 *** | −0.108 | 0.037 | −0.189 ** | −0.107 | 0.037 | −0.187 ** |
| Berthing time | | | | −0.155 | 0.053 | −0.181 ** | −0.146 | 0.053 | −0.171 ** |
| Parking price | | | | 0.154 | 0.057 | 0.165 ** | 0.156 | 0.056 | 0.167 ** |
| Distance to destination | | | | −0.085 | 0.037 | −0.139 * | −0.059 | 0.038 | −0.097 |
| Smart parking coupons | | | | | | | 0.122 | 0.058 | 0.125 * |
| Free parking fee for limited time | | | | | | | 0.068 | 0.041 | 0.102 |
| $R^2$ | | 0.209 | | | 0.303 | | | 0.331 | |
| $R^2$ change | | 0.19 | | | 0.277 | | | 0.298 | |

\* *p* < 0.05, \*\* *p* < 0.01, \*\*\* *p* < 0.01.

### 4.2. Investigating Psychological Factors in SPSs Adoption

After delving into the impact of objective elements on travelers' parking behavior using HRM, we now shift our focus to an equally integral component in deciphering the decision-making process inherent in the uptake of SPSs: the psychological determinants. This section deploys SEM, a potent analytical tool employed to dissect the complex network of interrelationships spanning attitude, perceived usefulness, perceived ease of use, social influence, and privacy concerns, all of which culminate in parking intention. Our model integrates a rich constellation of variables: intention (IT), attitude (AD), perceived usefulness (PU), perceived ease of use (PE), social influence (SI), and perceived privacy (PP). The intricate interplay among these variables is vividly illustrated in Figure 5. In order to examine the relationships among the variables in this study, a structural equation model was constructed.

In these equations, β coefficients represent path weights, and e denotes error terms. The model formulation is as follows.

(i) Measurement Equation (1) linking the measurement indicators (survey items) to the latent factors:

$$\xi\_rn = \psi\_r\Lambda\_ln + \eta\_rn, \text{ for } n = 1, \ldots, N \text{ and } r = 1, \ldots, R \tag{1}$$

(ii) Structural Equation (2) relating the explanatory variables to the mediator variables:

$$\Lambda\_ln = \rho\_l\Xi\_li + \varepsilon\_ln, \text{ for } n = 1, \ldots, N \text{ and } l = 1, \ldots, L \tag{2}$$

(iii) Structural Equation (3) linking the mediator variables to the dependent variable(s):

$$\Upsilon\_in = \tau\_z\Lambda\_ln + \zeta\_in, \text{ for } n = 1, \ldots, N \text{ and } i = 1, \ldots, I \quad (3)$$

(iv) Structural Equation (4) linking the explanatory variables directly to the dependent variable(s):

$$\Upsilon\_in = \sigma\_i\Xi\_li + \kappa\_in, \text{ for } n = 1, \ldots, N \text{ and } i = 1, \ldots, I \quad (4)$$

where:

$\zeta\_rn$ denotes the value of an indicator *r* of the latent construct $\Lambda^*ln$ as perceived by respondent *n*;

$\Lambda \times ln$ represents the value of the latent construct l for respondent *n*;

$\Xi\_li$ are the explanatory variables;

$\Upsilon\_in$ is a vector of the dependent variable(s) (e.g., intention to use);

Error terms are expressed as $\eta\_rn$, $\varepsilon\_ln$, $\zeta\_in$, and $\kappa\_in$ and follow a normal distribution with the respective covariance matrices $\Sigma\eta$, $\Sigma\varepsilon$, $\Sigma\zeta$, and $\Sigma\kappa$. Parameters $\psi\_r$, $\rho\_l$, $\tau\_z$, and $\sigma\_i$ need to be estimated.

Here, R indicators translates into writing *R* measurement equations and estimating a $(R \times 1)$ vector $\psi$ of parameters (i.e., one parameter is estimated for each equation), while *L* latent constructs translates into writing *L* structural equations and estimating an $(M \times L)$ matrix of $\rho$ parameters (i.e., *M* parameters are estimated for each equation).

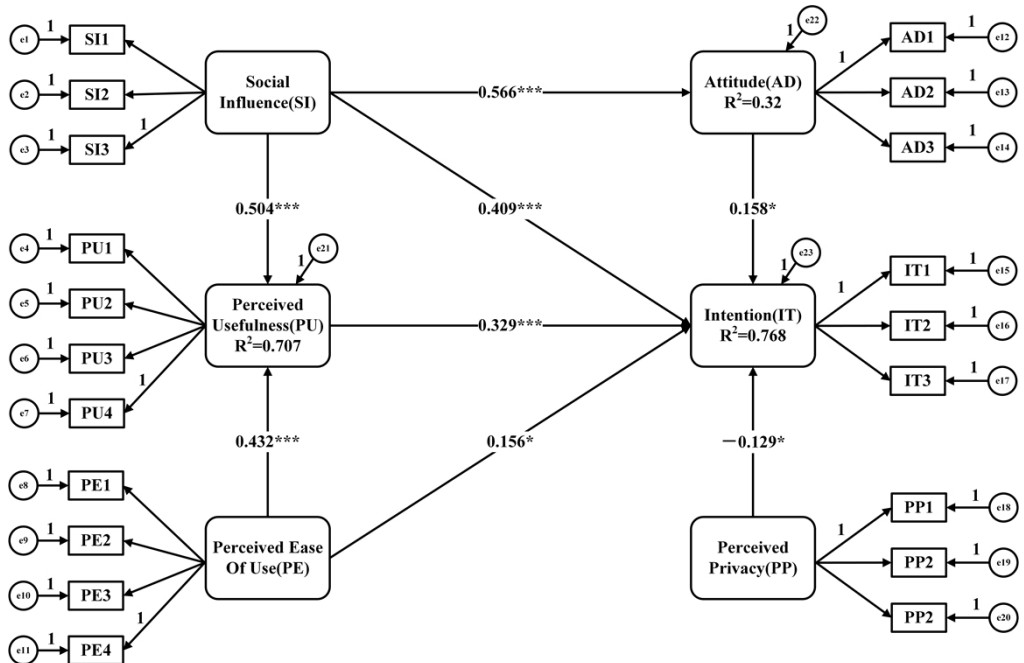

**Figure 5.** Standardized estimates for measurement and structural models. * $p < 0.05$, *** $p < 0.01$.

### 4.2.1. Goodness of Fit and Estimated Results

The goodness of fit and estimated results for the measurement model were as follows: the CMIN (minimum discrepancy) was 215.252 with 159 degrees of freedom (DFs), and the CMIN/DF ratio was 1.354, falling within the recommended range of 1 to 3, signifying an excellent fit. The comparative fit index (CFI) stood at 0.981, exceeding the recommended threshold of 0.95 and also indicating an excellent fit. The root mean square error of approximation (RMSEA) was 0.040, falling below the recommended threshold of 0.06 and suggesting an excellent fit. The Tucker–Lewis index (TLI) was 0.978, surpassing the recommended threshold of 0.95, which implies an excellent fit. In conclusion, the estimated results demonstrate that the model possessed excellent goodness of fit, fulfilling the criteria for a suitable model. Detailed results are shown in Table 4.

**Table 4.** Goodness of fit indices.

| Measure | Estimate | Threshold | Interpretation |
|---|---|---|---|
| CMIN | 215.252 | -- | -- |
| DF | 159 | -- | -- |
| CMIN/DF | 1.354 | Between 1 and 3 | Excellent |
| CFI | 0.981 | >0.95 | Excellent |
| RMSEA | 0.04 | <0.06 | Excellent |
| TLI | 0.978 | >0.05 | Excellent |

### 4.2.2. Hypothesis Testing

The final model's overall fit index indicated a satisfactory fit with the data, and the model was deemed acceptable. Consequently, the standardized path coefficients between latent variables could be employed to test the hypothesized relationships. The eight hypothetical relationships' test results are summarized in Table 5. To evaluate the hypotheses, the standardized regression coefficients were scrutinized. The hypothesis testing results are presented in Figure 5.

The overall goodness of fit index demonstrated that the final model fit the data satisfactorily, and it was thus accepted. Standardized path coefficients were utilized to examine the conceptual framework and assess the relationships between the variables of interest. A summary of the results is provided in Figure 5 and Table 5.

The findings supported H1, indicating that a more favorable attitude towards smart parking systems ($\beta1 = 0.178$, $p1 = 0.011$) positively influences travelers' intentions to utilize them. H2 was also confirmed, indicating that travelers' perception of the usefulness of smart parking systems ($\beta2 = 0.329$, $p2 < 0.001$) positively impacts their intentions to adopt them. Similarly, H3 was validated, suggesting that travelers' perception of the ease of use of smart parking systems ($\beta3 = 0.156$, $p3 = 0.029$) positively influences their intentions to use them. H4 was statistically significant, revealing that perceived ease of use ($\beta4 = 0.432$, $p4 < 0.001$) positively impacts perceived usefulness.

Moreover, H5 was supported, showing that social influence ($\beta5 = 0.409$, $p5 < 0.001$) exerts a significant positive effect on travelers' intentions to use smart parking systems. H6 was also validated, indicating that social influence ($\beta6 = 0.566$, $p6 < 0.001$) positively influences travelers' attitudes towards smart parking systems. H7 demonstrated that travelers' perceived privacy concerns ($\beta7 = -0.129$, $p7 = 0.015$) negatively affect their intentions to use smart parking systems. Lastly, H8 was supported, suggesting that social influence ($\beta8 = 0.504$, $p8 < 0.001$) positively impacts the perceived usefulness of smart parking systems.

**Table 5.** Hypothesis testing results.

| Hypothesis | Path Relationship | | | Standardized Estimate ($\beta$) | C.R. | $p$ |
|---|---|---|---|---|---|---|
| H1 | Intention | $\leftarrow$ | Attitude | 0.178 | 2.551 | 0.011 |
| H2 | Intention | $\leftarrow$ | Perceived usefulness | 0.329 | 3.509 | *** |
| H3 | Intention | $\leftarrow$ | Perceived ease of use | 0.156 | 2.178 | 0.029 |
| H4 | Perceived usefulness | $\leftarrow$ | Perceived ease of use | 0.429 | 6.119 | *** |
| H5 | Intention | $\leftarrow$ | Social influence | 0.409 | 4.593 | *** |
| H6 | Attitude | $\leftarrow$ | Social influence | 0.566 | 7.342 | *** |
| H7 | Intention | $\leftarrow$ | Perceived privacy | −0.129 | −2.443 | 0.015 |
| H8 | Perceived usefulness | $\leftarrow$ | Social influence | 0.504 | 6.831 | *** |

Note: *** indicates a significance level of 0.1%.

### 4.2.3. Analysis of Direct, Indirect, and Total Effects

The analysis results displayed in Figure 6 reveal several crucial insights. First, both attitude and perceived usefulness had direct positive effects on intention, suggesting that individuals are more likely to adopt intelligent parking systems if they have a favorable attitude towards them and perceive their utility. Moreover, perceived ease of use significantly influenced intention, displaying both direct effects and indirect effects mediated by social influence. This implies that the system's perceived ease of use is a pivotal factor in individuals' decisions to utilize it and that social influence contributes to shaping their perception of its user-friendliness. Additionally, social influence had a considerable impact on intention, with a direct effect of 0.409 and an indirect effect through attitude of 0.256, resulting in a total effect of 0.665. This indicates that social influence is a critical determinant in shaping individuals' intentions to use the system and that the opinions and behaviors of others significantly affect their decision-making process. Furthermore, social influence directly affected attitude with a path coefficient of 0.566, suggesting that others' attitudes towards the system can also influence an individual's own attitude. Conversely, perceived privacy concerns had a negative direct effect on intention with a path coefficient of −0.129. This implies that privacy concerns may discourage individuals from adopting intelligent parking systems. Finally, perceived usefulness directly and positively impacted social influence with a path coefficient of 0.504, signifying that individuals are more likely to recommend the system to others if they perceive it as useful.

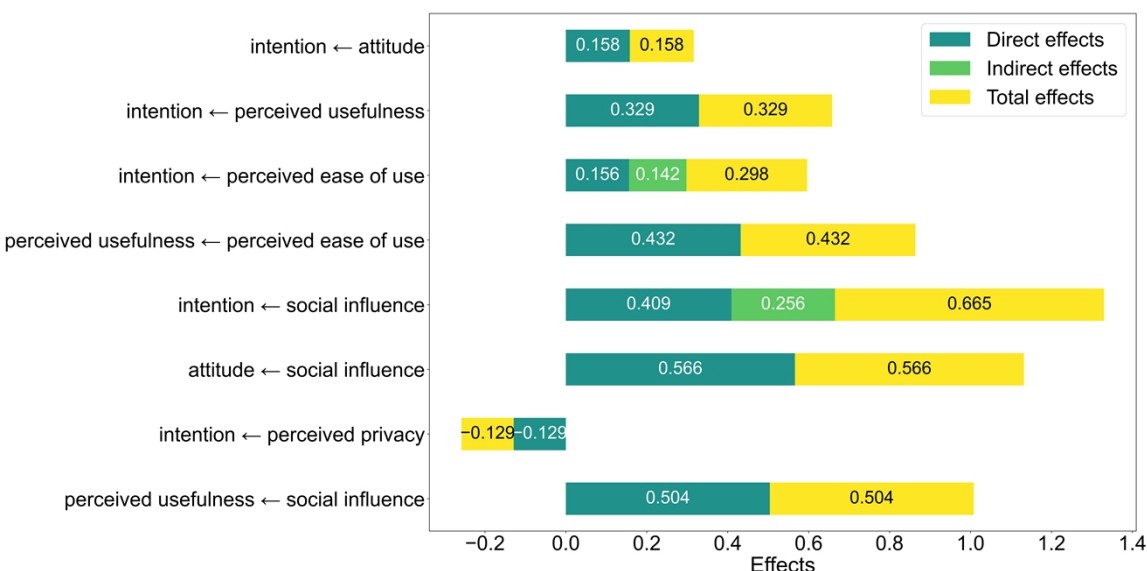

**Figure 6.** Comprehensive illustration of variable impacts on the intention to utilize SPSs.

### 5. Conclusions and Limitations

In this study, we endeavored to unravel the intricate determinants that shape travelers' parking behavior intentions with a specific focus on the impact these factors have on the adoption of SPSs. To comprehensively understand these multifaceted influences, we utilized an advanced, interconnected experimental methodology that integrated SEM and HRM.

The findings from our investigation illuminated several significant insights. Within the realm of psychological factors, social influence emerged as the most prominent determinant shaping travelers' parking behavior intentions, a deviation from previous studies where perceived usefulness or perceived ease of use often came to the forefront [48]. This was followed by perceived usefulness, perceived ease of use, and attitude. Interestingly, perceived privacy was the least influential factor among the psychological determinants. However, its unique and significant role in influencing behavior intentions cannot be understated given the growing emphasis on personal data security. This underlines the urgent necessity

for SPS operators to institute stringent privacy protection measures, thereby alleviating user concerns and fostering trust.

In terms of objective determinants, driving experience surfaced as the most influential factor, a notable contrast to earlier studies that prioritized factors such as parking prices, or berthing time [26]. This was followed by the presence of children, berthing time, parking price, education, and the availability of parking coupons. The variations in intentions to use SPSs among different demographic groups, particularly among individuals with children, emphasize the need for tailored strategies that cater to the diverse needs of these groups.

While our research provides knowledge on the determinants of travelers' SPS parking behavior intentions, it is important to note its limitations and the potential avenues for future research. The study was conducted in a specific geographical and cultural context, and the extent to which these findings can be generalized to other regions remains a question. Future research could broaden the scope of this study by incorporating a diverse range of contexts and demographic characteristics to increase the generalizability of the results.

**Author Contributions:** Methodology, Y.Z.; Investigation, Y.Z. and X.S.; Writing—original draft, Y.Z.; Writing—review and editing, Y.Z. and P.T.; Supervision, H.L., T.Z. and Q.C. All authors have read and agreed to the published version of the manuscript.

**Funding:** This study was supported by the National Natural Science Foundation of China (no. 52131202).

**Institutional Review Board Statement:** Not applicable.

**Informed Consent Statement:** Informed consent was obtained from all subjects involved in the study.

**Data Availability Statement:** The data are available upon request.

**Conflicts of Interest:** The authors declare no conflict of interest.

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
