# Peer review of "Investigating Factors for Travelers’ Parking Behavior Intentions in Changchun, China, under the Influence of Smart Parking Systems"

_sustainability, doi:10.3390/su151511685_

Round 1

Reviewer 1 Report

This paper explains about “Investigating factors for travelers' parking behavior intention in Changchun, China under the influence of smart parking systems”. The topic of the paper draws interest. However, the reviewer views the paper needs a significant effort to improve its quality to ensure publication in the journal. The reviewer recommends further major revision of the paper. Below are some comments to help improve the paper:

The reviewer suggestion to provide the highlight to give understand about this study.

The abstract should be written in concise, consists of background, reason or objectives of the study, methodology, focus on the key points of the results and writing in one paragraph. Please give attention about it.

increased the quality of the figures 2, 4 and 6.

·         Regarding this you can refer the paper and provide the proper citation.

The reviewer suggested that the findings of the engine be included in order to provide a better understanding of this investigation.

The novelty and objective can be re-written and to justify the uniqueness of your work.

Please state comparison of your findings with other published papers or works, so that your manuscript could contribute to the new knowledge

The introduction discusses clearly the importance of smart parking

Low, major revision.

Reviewer 2 Report

Dear authors

The topic is interest. The study is completed with conducting a meaningful face-to-face survey. However, the manuscript is hard to read and comprehend due to  the vague logic to explain the significance of the present study and writting issues. Authors tended to investigate PSYCHOLOGICAL FACTORS, but the manuscript was not written in a good PSYCHOLOGICAL way, that is lack of the rigorously scientific sense.

1. Motivation is vague. Although authors mentioned that "understanding the determinants of travelers' parking behavior intentions is essential for promoting the daoption of SPS and forstering sustainable urban mobility", it is too general to  comprehend the significance of the present study.

2. Since the abbreviations (such as SPS, TPB TAM) were defined, please use them after the difinitions. However, there are lots of difinitions repeatedly, which degraded the readability of the manuscript.

3. Since the survey was conducted within Changchun's smart city, it is necessary to consider the traffic density, vehicle's prevalence, averaged income of the city, which might be included in (iii) situational and promtional factors. This is related to the validness of catagorizing and specifying those possible factors in a sysmetic way. Although authors tended to use some theoretical models such as TPB, TAM, and PCT to discuss safety, privacy, attitude and social influence, there is no rational description to explain why and only these contents were chosen. Hence, the unclear issue led to question the significance of the study.

4. Is it schematic representaion of intention int the context of SPS proposed originally or expanded based any theoretical model? There is no any citation in Section 3.1 to explain the framework, and also no definite descript to indicate that the framework was proposed originally. Please explain it.

5.  Concept of "psycologyical factors" is really vague due to lack of theorectic background. No definition of  "psycologyical factors" was given, and also no explanation why they used these "psycologyical factors" was given. That is, why did authors discuss those "psycologyical factors" of attitude, perceived usefulness and perceived ease of use, social influence, privacy concerns at some level. (that is also related to 4.)

6. Paragraph (line 198-201) mentioned Fig. 1 and Table. Why? There was no explanation about the data source. It is too confusing!

7. Please untify the digit after decimal point. For example 54.79% (line 238) , 83.1% (line 239), 0.129, 0.015 (page12).

8. the sentence in line 232-233 is repeated with that in line 223-234.

9. The manuscript lacked the further discussion, and limitations.

I strongly recommend authors to check the manuscript in a whole, not only language, but also constructure.

Reviewer 3 Report

- The research work's contributions should be clearly outlined.

- Authors should provide additional results and analysis.

-The authors should make the figures more visible. The majority of the figures in the article have low resolutions.

- The contents of the conclusion appear to be very extensive. It must be decreased.

The authors must proofread the entire article to ensure proper English usage.

Round 2

Reviewer 1 Report

Accepted.

Quality of English Good.

Reviewer 3 Report

The article has merit to be accepted in the present form.

A proofreading required for minor spelling and grammar check.